# Alveolar Type 2 Epithelial Cell Organoids: Focus on Culture Methods

**DOI:** 10.3390/biomedicines11113034

**Published:** 2023-11-12

**Authors:** Krishan Gopal Jain, Nan Miles Xi, Runzhen Zhao, Waqas Ahmad, Gibran Ali, Hong-Long Ji

**Affiliations:** 1Department of Surgery, Health Sciences Division, Stritch School of Medicine, Loyola University Chicago, Maywood, IL 60153, USA; kjain2@luc.edu (K.G.J.); rzhao6@luc.edu (R.Z.); wahmad1@luc.edu (W.A.); 2Burn and Shock Trauma Research Institute, Health Sciences Division, Stritch School of Medicine, Loyola University Chicago, Maywood, IL 60153, USA; 3Department of Mathematics and Statistics, Loyola University Chicago, Chicago, IL 60660, USA; mxi1@luc.edu; 4Division of Pulmonary and Critical Care Medicine, Mayo Clinic, Rochester, MN 55905, USA; ali.gibran@mayo.edu

**Keywords:** organoids, alveolar type 2 cells, stem cells, 3D cultures, pulmonary diseases

## Abstract

Lung diseases rank third in terms of mortality and represent a significant economic burden globally. Scientists have been conducting research to better understand respiratory diseases and find treatments for them. An ideal in vitro model must mimic the in vivo organ structure, physiology, and pathology. Organoids are self-organizing, three-dimensional (3D) structures originating from adult stem cells, embryonic lung bud progenitors, embryonic stem cells (ESCs), and induced pluripotent stem cells (iPSCs). These 3D organoid cultures may provide a platform for exploring tissue development, the regulatory mechanisms related to the repair of lung epithelia, pathophysiological and immunomodulatory responses to different respiratory conditions, and screening compounds for new drugs. To create 3D lung organoids in vitro, both co-culture and feeder-free methods have been used. However, there exists substantial heterogeneity in the organoid culture methods, including the sources of AT2 cells, media composition, and feeder cell origins. This article highlights the currently available methods for growing AT2 organoids and prospective improvements to improve the available culture techniques/conditions. Further, we discuss various applications, particularly those aimed at modeling human distal lung diseases and cell therapy.

## 1. Introduction

Alveoli, located at the distal end of the lung, are tiny air sacs whose epithelial layer consists of two types of epithelial cells [1]. Type 1 epithelial (AT1) cells cover approximately 95% of the alveolar surface area and are responsible for gaseous exchange. Type 2 alveolar cells (AT2) form the remaining 5% of the alveolar surface area and are responsible for surfactant secretion, immune defense against infections, and the regulation of fluid volume. AT2 cells serve as the facultative adult stem cells of the distal lung, and they are responsible for regenerating the alveoli through self-renewal and differentiation into AT1 cells [2,3,4]. Distal lung diseases, such as COPD/emphysema, and infections including severe acute respiratory syndrome (SARS) and COVID-19, can cause damage to the alveolar epithelium and are among the leading causes of global mortality [5,6,7,8].

Studies on primary and secondary respiratory diseases have been limited by the scarcity of models for the human alveolar epithelium that faithfully mimic in vivo physiology and pathophysiology [9]. For decades, the 2D air–liquid interface model of AT2 cells has been the most extensively studied tool for modeling epithelial barrier function and lung diseases [10,11]. While this 2D model has provided valuable insights into the alveolar structure and function of distal lungs, enriching our knowledge of alveolar diseases, it has often failed to maintain cell proliferation and the expression of tissue-specific functions [12]. Typically, 2D AT2 cell models require primary AT2 cells isolated from humans and animals, with only a few research groups successfully cultivating polarized mouse AT2 monolayers [13,14,15,16,17]. In contrast, 3D model culture systems are far superior in mimicking in vivo lung conditions [18]. Over the past decade, 3D organoid models have been extensively used to study lung diseases. Organoids are self-organized, three-dimensional cultures derived from stem cells in vitro, closely recapitulating the natural structural organization and physiological conditions of organs [19].

The term “organoid” was initially used in 1946 to describe the histological features of tumors [20]. Later, this term came to refer to tissues or structures that resemble an organ in vitro (Figure 1). These structures are often called miniature organs due to their ability to recapitulate the complex structure and function of the corresponding organs. After six decades, in 2014, Lancaster et al. defined organoids as a 3D self-organizing structure [21]. The first 3D organoids were “enteroids” developed from mouse intestinal stem cells [22]. Advancements in comprehending the techniques and identifying the necessary growth factors for organoid culture have led to the establishment of organoids from a range of tissues, such as the salivary gland, midbrain, colon, pancreas, retina, liver, prostate, and lung, in multiple laboratory settings [23,24,25,26,27,28,29]. The initial lung organoids were airway organoids pioneered by Rock et al. [30]. Shortly thereafter, McQuarter et al. developed a mixed culture of airway and alveolar organoids [31]. In 2013, Barkauskas et al. made a significant advancement by successfully developing alveolar organoids from both human and mouse AT2 cells [4]. Concurrently, the progress in stem cell research sparked interest among scientists in utilizing embryonic stem cells (ESCs) and induced pluripotent stem cells (iPSCs) as a consistent cell source for cultivating organoids [32]. Nikolic et al. harnessed multipotent human embryonic lung bud tip epithelial progenitor cells to grow lung organoids [33]. Initially, organoid cultures relied on feeder cells, but later, chemically defined media replaced the need for feeder cells in 3D organoid cultures [3,34,35]. Feeder-free alveolar organoids offer an ideal platform for the in vitro expansion of AT2 cells, a task that was previously challenging in 2D cultures due to their tendency to spontaneously differentiate into AT1 cells. These 3D organoid models can be employed for the proliferation and expansion of AT2 cells for months. Recently, these feeder-free organoid-derived AT2 cells have been used in mimicking the lung-on-chip model [36]. These organoids mimic both the structure and functions of the alveoli. Recently, Lim et al. [37] showed that NKX2.1 regulates alveolar differentiation and functional maturation like the presence of lamellar bodies and the production and secretion of surfactants in human fetal lung AT2-derived organoids. Organoid technology is increasingly employed to model animal and human organ development, as well as various human pathologies in a dish.

Mouse lungs are not ideal models for investigating human lung diseases due to the significant differences between them and human lungs in various aspects. These distinctions include size, lung structure, cellular composition, the absence of cartilaginous rings, respiratory bronchioles, and cytokeratin 5+ basal stem cells in mouse lungs [38]. Furthermore, the developmental stages of mouse lungs differ from those of human lungs in terms of molecular pathways, metabolic processes, and the absence of the ACE2 receptor [39,40]. Therefore, it is of utmost importance to identify a model that closely mimics the human lung in vitro. Three-dimensional organoids derived from human AT2 cells play a crucial role in bridging the gaps between human and mouse lungs, offering a robust platform for studying human lung pathologies, lung development, and the development of new therapeutic approaches [41]. Additionally, 3D AT2 organoids hold the potential to complement whole lung transplantation in the field of regenerative medicine. Laboratories lacking access to human tissue can utilize humanized mice derived AT2 cells to cultivate these organoids [42]. AT2 organoids also offer flexibility in selecting specific cell types of interest, a feature not possible with in vivo mouse models. AT2 alveolar organoids can be cultivated as either epithelial organoids or co-culture organoids, involving AT2 cells in combination with endothelial and/or mesenchymal cells, depending on the specific research requirements [43].

Given their miniature organ properties, organoids offer a valuable alternative to reduce, refine, and replace (3R) animal experiments, addressing ethical issues. Moreover, data generated from transformed or cancer cell lines and animal models may not be effectively translated into clinical applications due to genetic abnormalities and differences in genetic makeup [44,45]. Human-tissue-derived organoids currently represent the best model for faithfully mimicking cell composition, tissue organization, and physiological functions akin to natural organs within the body. Findings from human tissue models can be directly applied in clinical contexts.

This review focuses on various methods used to grow AT2 organoids and their applications in studying lung development, lung diseases, and designing new therapies. We discuss a possible advancement that has the potential to generate new rapid disease screening tools, disease modeling, and future regenerative strategies. Additionally, we provide an overview of the limitations of this organoid model.

## 2. Literature Search Strategy

The first study on pure alveolar organoids was published in 2013 [4]. Therefore, we conducted searches on PubMed and Google Scholar to identify the literature published from January 2013 to October 2023 using the following keywords: “AT2 epithelial cell organoid”, “alveolar type 2 epithelial cell organoid” or “alveolar organoid”. We screened articles focused on alveolar organoid cultures and their applications. Then, we categorized studies based on the sources of cells, matrix, culture modes (i.e., feeder-free or co-culture), growth factors, and medium used. After screening the titles and abstracts of relevant publications, we included the full texts of the selected publications for review. Furthermore, we identified a few additional studies by examining the references cited in the selected articles.

## 3. Sources of AT2 Cells

Adult lungs consist of more than 40 types of cells, with AT2 cells comprising about 90% of the alveolar epithelial cell population in the distal lung [12,46]. Various strategies have been employed to isolate these cells for alveolosphere cultures. The distal portion, representing the alveolar region, has been collected and subjected to enzymatic digestion. This process allowed for the collection of a cell mixture comprising various cell types, including stromal cells, the capillary endothelium, pericytes, and macrophages. AT2 cells have been harvested with a purity of 90–95% using the “panning method”, in which IgG-coated plates are used to attach AT2 cells [47]. Others have also reported using the density gradient centrifugation method to isolate AT2 cells with 90% purity [48]. Other sophisticated methods include genetic lineage labeling (Sftpc-CreERT2; Rosa-tdTomato) in vivo or surface marker labeling in vitro (EpCAM, HTII280). Subsequently, lineage-positive or surface-marker-positive cells have been purified using MACS (magnetic cell sorting), FACS (fluorescent cell sorting), or a combination of both methods [4,31,49,50]. Furthermore, research studies have noted the presence of distinct subsets of AT2 cells within the alveoli that display variations in their proliferative potential and exhibit unique responses during both homeostasis and injury [1]. In an interesting study, Hasegawa et al. revealed that relying solely on the EpCAM^+^ marker is insufficient for the purification of AT2 cells from a mixed lung cell population [51,52]. By introducing major histocompatibility complex class II (MHCII) as a secondary marker, they demonstrated that the FACS-sorted EpCAM^+^ epithelial cells comprised three distinct subpopulations. Subpopulation 1 consisted of EpCAM^med^ MHCII^+^ cells enriched in proSP-C^+^ expressing cells. Subpopulation 2, characterized by EpCAM^hi^ MHCII^−^ expression, primarily consisted of ciliated cells. Subpopulation 3, marked by EpCAM^low^ MHCII^−^ expression, comprised AT1 cells. Notably, the expression levels of EpCAM and MHCII remained largely unchanged across different mouse strains and ages or in response to lipopolysaccharide (LPS)-induced lung injury [51]. Embryonic bipotent alveolar epithelial progenitors, specifically at the e16.5 stage, have been employed as a valuable source of 3D organoid cultures as well [2]. To enhance the yield of bipotent alveolar epithelial progenitors, multiple embryos have been combined from the same litter, typically five to seven embryos. Nonetheless, the isolation of primary AT2 cells from both adult and embryonic lungs is a labor-intensive and intricate procedure. More recently, researchers have turned to pluripotent stem cells (PSCs), including both iPSC-derived and ESC-derived AT2 (iAT2) cells, as an alternative approach to cultivating alveolar organoids. This method offers advantages in terms of efficiency and reproducibility [19,53,54]. Thanks to their unlimited proliferation potential in vitro, PSCs could serve as a continuous source of iAT2 cells and contribute to animal welfare. To derive iAT2, pluripotent stem cells have been directed to differentiate into distal lung SFTPC^+^ alveolar epithelial cells using a multistep protocol. A well-defined serum and feeder-free combination of lineage-inducing factors recreated the in vivo development niche, initially inducing pluripotent cells into definitive endoderm and subsequently patterning them into anterior foregut lineage cells. These cells were then differentiated into primordial NKX2-1^+^ lung progenitor cells and finally into distal lung SFTPC^+^ alveolar epithelial cells [55,56,57].

## 4. AT2 Organoid Culture Methods

Lung AT2 organoids serve as a robust model for studying lung diseases, drug screening, investigating lung development, and designing new therapies (Figure 2). Over past decades, various methods have been employed to cultivate distal lung organoids, depending on the cell sources, growth supplements, and supporting cells. The most utilized approaches include co-culture and feeder-free (AT2-cell-only) procedures.

### 4.1. Co-Culture Method

Three-dimensional in vitro organoid cultures rely on the components present in the microenvironment of cells in vivo (Table 1). Fibroblast niches regulate AT2 cells’ self-renewal or trans-differentiation into AT1 via Wnt signaling both in vitro and in vivo [58]. These paracrine signals from fibroblast niches are crucial in determining the destiny of AT2 cells. To grow AT2 organoids and replicate their function in vitro, feeder cells serve as a source of niche signals when co-cultured with AT2 cells. Various supporting cell types, including cell lines [43], primary lung fibroblasts [59], PDGFRa^+^ fibroblasts or endothelial cells [60], MSCs [31], or PSC-derived lung mesenchymal cells, have been used with AT2 cells to support organoid growth [61]. However, these supporting cells not only secrete signals for AT2 cells but also secrete other components required for their own growth. The factors secreted from dying fibroblast or endothelial cells may also affect the function of AT2 cells, which could impact our research question. Therefore, co-culture is not a controlled system for studying the molecular mechanism of drug action.

To determine the growth factor requirements for AT2 organoid colony formation, co-cultures are supplemented with a medium consisting of various cytokines known to be involved in lung development. McQualter et al. demonstrated that epithelial growth factors like fibroblast growth factor (FGF)-10 and/or hepatocyte growth factor (HGF) significantly increased organoid colony counts, but FGF-7 had no effect. In contrast, the addition of mesenchymal growth factors such as bone morphogenic protein 4 (BMP-4), TGF-β1, or platelet-derived growth factor (PDGF) either significantly reduced or completely inhibited organoid colony formation [31]. Besides, FGF-10 or HGF supported EpCAM^+^ organoid formation in mesenchymal-free Matrigel cultures. When EpCAM^+^ cells were treated with a combination of FGF-10 and HGF, an increase in colony size and colony formation efficiency was observed, indicating that these growth factors acted synergistically to support organoid proliferation [62]. However, culture media containing fetal bovine serum (FBS) are poorly defined and contain unknown factors [63]. Some of these factors may not be required for AT2 sphere cultures at all but can affect AT2 phenotype and functions. Such complex conditions do not provide a modular system in which AT2s can be either selectively expanded or differentiated into AT1 [34,64].

**Table 1 biomedicines-11-03034-t001:** Organoid culture from primary AT2 cells.

Cell Source	Feeder Cell Type (Epithelial-to-Feeder Ratio)	Medium	Model
Mouse AT2			
Genetic-lineage-labeled AT2 cells [4]	PDGFRα^+^ lung lipofibroblasts (1:20)	MTEC^+^: DMEM/F12, ITS, EGF, BPE, CTX, 5% FBS, and antibiotics; RA freshly added; Y-27632 for the first 2 days	The first organoid model to show that AT2 cells are stem cells of the adult distal lung
EpCAM^+^ AT2 cells [31]	Lung mesenchymal cells (EpCAM^−^ Sca1^+^) (1:1)	DMEM/F12 plus L-glutamine with 10% newborn calf serum, ITS, FGF-7, FGF-10, penicillin/streptomycin, HGF, BMP-4, TGF-, and PDGF-AA	Epithelial stem/progenitor cell hierarchy in mouse lung
Axin2^+^ or Sftpc^+^ AT2 cells [65]	Adult primary lung fibroblasts (1:10)	MTEC-SAGM with Wnt3a	Wnt signaling regulates lung alveologenesis
EpCAM^+^ AT2 cells [60]	Lung endothelial cells (LUMECs) (1:50,000)	DMEM/F12 with ITS, 10% FBS, 1 mM HEPES pH 7.5, glutamine, and antibiotics	AT2 lineage regulation via endothelial cells
EpCAM (CD326)^+^ [66]	AXIN2^+^ mesenchymal alveolar niche cells (1:10)	MTEC plus media (DMEM/Ham’s F-12, HEPES, penicillin and streptomycin, fungizone, insulin, transferrin, cholera toxin, EGF, and bovine pituitary extract)	Effect of disease or aged fibroblast on organoid
Plau^−/−^ AT2, Scnn1D Tg AT2 [16,42]	Mlg2980 (3:100)	DMEM/F12 with ITS, 10% FBS, glutamine, antibiotics, and SB431542	AT2 lineage regulation via fibrinolytic niche and dENaC
PAI-1 Tg AT2 [43]	Mlg2980 (1:50) and feeder-free	DMEM/F12, penicillin–streptomycin, CHIR, DMH, BIRb 1, rmsNoggin, rmsFGF-10, heparin, rhEGF, Y-27632, dihydrochloride, B27 supplement, glutaMAX, and HEPES	AT2 lineage regulation via fibrinolytic niche
FACS sorted AT2 cells [3,67]	Feeder-free	DMEM/F12, penicillin–streptomycin, CHIR, DMH, BIRb 1, rmsNoggin, rmsFGF-10, heparin, rhEGF, Y-27632, dihydrochloride, B27 supplement, glutaMAX, and HEPES	Method for FF alveolar organoid
CD45^−^ EpCAM^+^ β4^−^ AT2 [34]	Feeder-free	5% charcoal-treated FBS, penicillin–streptomycin, SABMTM Basal Medium + SAGMTM Supplement Pack, A83-01, ms Wnt3a, msNoggin, h FGF-10, hKGF (FGF-7), msEGF, hRspondin-1, Y-27632, dihydrochloride, B27 supplement, glutaMAX, HEPES, and N2 supplement	Transplantation and cell therapy after lung injury
EpCAM^+^ AT2 cells [58]	Feeder-free	DMEM/F12, 1%FBS, and Fgf-7	Single-cell Wnt signaling niche
CD31-CD45-EPCAM^+^ Sca1^−^ cells [68]	Neonatal stromal cells (1:10)	DMEM/F12 with 10% FBS, penicillin/streptomycin, 1 mM HEPES, and insulin/transferrin/selenium (Corning) (3D media)	Early-stage lung adenocarcinoma
Human AT2			
EpCAM^+^and HTII-280^+^ AT2 [4]	MRC5 fibroblasts (Not reported)	ALI medium [69]	The first organoid model to show that AT2 cells are stem cells of the adult distal lung
EpCAM^+^, HTII-280^+^, and TM4SF1^+^ AT2 [62]	MRC5 fibroblasts (1:10)	Small Airway Growth medium, Y27632, Wnt3a, FGF-7, FGF-10, XAV939, and CHIR99021	Lung regeneration
AT2 cells [9,67]	Feeder-free	Chemically defined EGF/Noggin medium	SARS-CoV-2 infection and COVID-19-associated pneumonia
MACS sorted AT2 cells [3]	Feeder-free	DMEM/F12, penicillin–streptomycin, CHIR, DMH, BIRb 1, rmsNoggin, rmsFGF-10, heparin, rhEGF, Y-27632, dihydrochloride, B27 supplement, glutaMAX, and HEPES	Method for FF alveolar organoid

#### 4.1.1. EpCAM^+^ AT2 Co-Cultured with Fibroblast Cells

Earlier attempts to grow organoids in Matrigel from EpCAM^+^ AT2 cells alone failed, but complex epithelial cell colonies were generated when co-cultured with Sca-1^+^ mesenchymal cells [31]. This study showed that αSMA^+^ mesenchymal cells tightly surrounded epithelial colonies, suggesting a dependency of lung epithelial cells on feeder (fibroblast)-cell-released FGF-10 and HGF to form colonies in vitro. This observation aligns with in vivo studies demonstrating that FGF-10 and HGF regulate lung development [46,70,71]. Later, McQualter et al. [72] showed that CD166^−^ and CD166^+^ lung stromal cells exhibited different epithelial-supportive capacities. They reported that the in vitro expansion of lung stromal cells resulted in the downregulation of FGF10 expression, reducing their ability to support epithelial colony formation. They also found that the pretreatment of stromal cells with TGF-β1 reduced FGF-10 expression and their capacity to support epithelial colony formation. Conversely, pretreatment with a TGF-β1 inhibitor (SB431542) upregulated FGF-10 expression and enhanced their capacity to support epithelial organoid formation. These findings suggest that TGF-β1 regulates the epithelial supportive capacity of lung stromal cells via FGF-10 signaling. Additionally, studies have revealed that FGF-10 expression is decreased in the lungs of patients with bronchopulmonary dysplasia (BPD), and FGF-10 treatment can induce de novo alveologenesis in the lungs of BPD mice [73].

#### 4.1.2. AT2 Co-Cultured with PDGFRA^+^ Cells

Barkauskas et al. reported that AT2 cells function as stem cells in the distal lung and form alveolospheres in vitro [4]. They genetically labeled AT2 cells and purified Sftpc-CreER:Rosa-Tm lineage-positive cells using FACS. To validate that PDGFRA^+^ mesenchyme cells were an integral part of AT2 niches, these AT2 populations were co-cultured with primary PDGFRA^+^ mesenchyme cells (in a 1:20 ratio) or with neonatal mouse lung fibroblasts. They found that AT2 cells co-cultured with PDGFRA^+^ mesenchyme cells developed larger and more numerous colonies compared to those co-cultured with immortalized fibroblast cell lines. Furthermore, the same group reproduced their murine alveolar model in humans by growing HTII-280^+^ AT2 cells co-cultured with the MRC5 cell line to form alveolospheres. Subsequently, several studies adapted this protocol with variations in culture media composition, the AT2-to-fibroblast ratio, and the types of feeder cells used (see Table 1, Table 2 and Table 3). In general, AT2 cells are mixed with feeder cells in a DMEM/F12 medium supplemented with 10% FBS, amphotericin B penicillin/streptomycin, ITS, and heparin sodium salt. This cell suspension is then mixed with growth-factor-reduced Matrigel in a 1:1 ratio and either placed in transwell inserts or cultured as domes in petri plates, allowing them to grow for up to 14 days. It is important to note that lung mesenchyme niches have unique regulatory functions, with Pdgfrα-expressing mesenchyme cells supporting alveolar epithelial cell growth and self-renewal. In contrast, the Axin2^+^ myofibrogenic progenitor cells tend to generate pathologically deleterious myofibroblasts following injury [74].

#### 4.1.3. Human iPSC-Derived AT2

Induced pluripotent stem cells are adult cells genetically reprogrammed to resemble embryonic-like stem cells [75]. Patient-specific iPSCs can be a valuable tool for directly studying human diseases and developing personalized medicine using human-derived in vitro models (Table 2). Several studies have reported the differentiation of iPSCs into NKX2-1^+^ and SFTPC-expressing AT2 cells [57,76]. Recently, Jacob et al. reported a modified, robust protocol for the differentiation of iPSC into mature, functional AT2 comparable to primary adult AT2 cells. They found that Wnt signaling, in addition to FGF signaling, together with corticosteroids and cyclic AMP, promotes the maturation of SFTPC^+^ AT2 from NKX2-1^+^ precursors in vitro. The feeder-free alveolosphere derived from these AT2 cells displayed classical functional features of mature AT2 cells, including innate immune responsiveness and the processing of surfactant lamellar bodies. This iAT2-derived organoid model could be passaged for up to one year without differentiating into AT1 cells and serving as a source of pure AT2 population [77].

In an interesting study, patient-derived iPSCs were differentiated into NKX2-1^+^ lung epithelial progenitor cells through a 21-day differentiation method. The NKX2-1^+^ progenitor cells were sorted using Carboxypeptidase M (CPM) as their cell surface marker and cultured alone to grow lung bud organoids. Alternatively, these cells were co-cultured with primary lung fibroblasts to grow alveolar organoids to model HPS-associated interstitial pneumonia (HPSIP) [59]. However, this study could not show any significant differences in organoid morphology and size between patient-specific and gene-corrected organoids, likely due to the immaturity of iPSC-derived AT2 cells.

Yamamoto et al. reported a method for the long-term expansion of alveolar organoids [12,56] from human-induced pluripotent stem cells (hiPSC). These organoids contained hiPSC-derived SFTPC^+^ AT2 stem cells and were able to self-renew and differentiate into alveolar epithelial type I (AT1)-like cells. The transcriptomes and morphology of these cells were consistent with those of primary AT2 cells. Single-cell RNA-seq demonstrated that the differentiation process and cellular heterogeneity of iPSCs-derived AT2 cells resembled those of AT2 cells developing in vivo.

In a recent report, both AT2 and feeder cells derived from pluripotent stem cells were used to develop alveolar organoids to model influenza A (H1N1) virus and severe acute respiratory syndrome coronavirus 2 (SARS-CoV-2) infection [61]. We could find only one study that reported the alveolar organoid culture from mouse iPSCs. In this study, both epithelial and mesenchymal cells were derived from mouse iPSCs. The co-culture of iPSC-derived mesenchyme and lung epithelial progenitors gave rise to lung organoids composed of juxtaposed layers of separately derived engineered lung mesenchyme and engineered lung epithelium. Using this model, they identified that functional epithelial-mesenchymal crosstalk improved lung epithelial lineage specification [78].

**Table 2 biomedicines-11-03034-t002:** Organoid culture from iPSC-derived AT2 cells.

Cell Source	Feeder Cell Type (Epithelial-to-Feeder Ratio)	Medium	Model
Mouse iPSC			
iPSC-derived Nkx2-1^mCherry±^ AT2 cells [78]	iPSC-derived mesenchyme (1:20)	cSFDM (with RA-containing B27 supplement) rmWNT3A, rhFGF-2, rhFGF-10, and Y-27632	Epithelial–mesenchymal crosstalk in a multi-lineage lung organoid model
Human iPC			
iPSC-derived AT2 cells [68]	Feeder-free	CK+DCI+Y-27632 medium (cAMP/IBMX), CHIR, rhKGF, dexamethasone, and Y-27632 rho-associated kinase inhibitor	Early-stage lung adenocarcinoma model
hIPSC-derived CPM^+^ NKX2.1^+^ foregut endoderm cells [57]	Human fetal lung fibroblasts (1:50)	Medium consisted of RA, CHIR99021, and BMP4, FGF-10, dexamethasone, 8-Br-cAMP, 3-IBMX, and FGF-7	Carboxypeptidase M (CPM) as a surface marker of NKX2-1^+^ AEPCs
hIPSC-derived NKX2.1^+^ anterior foregut endoderm cells [56]	Fetal lung fibroblast (1:50) and feeder-free	FGF-7, FGF-10, dexamethasone, B27, ITS, 8-Br-cAMP, 3IBMX, CHIR-99021, BSA, CaCl2, Y-27632, and SB431542	Long-term culture model for drug toxicity screening
hIPSC-derived anterior foregut endoderm cells NKX2.1 GFP^+^/SFTPC tdTomato^+^ expression or CD47hi/CD26lo [55,77]	Feeder-free	FGF-7, FGF-10, dexamethasone, 8-Br-cAMP, 3IBMX, CHIR99021, and SB431542	Gene mutation correction model
hIPSc-derived NKX2.1^+^ progenitor cells [59]	Human fetal lung fibroblasts (1:50)	DCIK medium containing dexamethasone, 100 μM of 8-Br-cAMP, 100 μM of 3-isobutyl-1-methylxanthine, and 10 ng/mL of KGF	Hermansky–Pudlak syndrome type I modeling
hiPSC-derived GFP^+^ iAT2 cells [79]	Feeder-free	IMDM/Ham’s F12 media with CHIR99021, KGF, dexamethasone, 3-Isobutyl-1-methylxanthine (IBMX), 8-bromo-cAMP and primocin (CK+DCI medium), Y-27632, 5 days in K+DCI (without CHIR99021), and 7 days in CK+DCI	Matrigel-free synthetic hydrogel with microcavities for organoid culture

#### 4.1.4. Mouse and Human ESC-Derived AT2 Cells

Embryonic stem cells are pluripotent cells with the potential to differentiate into multiple tissue types (Table 3). Recently, bone-marrow-derived “very small embryonic-like stem cells” (VSELs) have been shown to have the potential to differentiate in vivo into SPC-producing AT2 cells. In a remarkable experiment, VSELs were isolated from SPC-H2B-GFP BAC transgenic mice and administered to a bleomycin-injured lung injury mouse model. After three weeks, VSELs differentiated into GFP^+^ AT2 cells and regenerated the epithelium in vivo. These VSELs-derived GFP^+^ AT2 cells were FACS-sorted from mice lungs and used to cultivate organoids in vitro to study their proliferation and differentiation potential. After 21 days of co-culture with MLG cells, AT2 cells from VSEL-transplanted mice developed GFP^+^ organoids, whereas no GFP^+^ organoids were observed in the control groups. The lineage of VSEL cells was confirmed via the colocalization of the TTF1 (transcription termination factor 1) marker with GFP^+^ AT2 cells [80].

Nikolić et al. utilized human embryonic lung bud tips progenitor cells to cultivate long-term self-renewing, branching organoids for studying lung development. The embryonic tips, embedded in Matrigel with EGF, FGF7, FGF10, Noggin, RSPO1, CHIR99021, and SB431542, formed spheres within 12 h with a 100% colony-forming efficiency. They were able to passage these organoids for nine passages without changes in morphology, SOX2 and SOX9 expression, or karyotype alterations. This study also showed that the growth conditions for human lung tip organoids do not support the long-term self-renewal of mouse lung tip cells, suggesting species-specific differences during lung development. Interestingly, they observed that mesenchymal cells disappeared after the second passage of organoid cultures, indicating that organoids can be maintained without mesenchymal cells. However, co-culture with canalicular-stage mesenchyme improved alveolar differentiation, leading to the attainment of a bipotent progenitor stage (pro-SFTPC^+^, HTII-280^+^, HOPX^+^ and PDPN^+^ co-expression, and NKX2-1^+^) [33,81].

Another study reported the development of alveolar organoids from hESCs via sequential differentiation into a definitive endoderm (DE), an anterior foregut endoderm (AFE), a ventral anterior foregut endoderm (VAFE), lung progenitors (LPs), and alveolar organoids. On day 21, these organoids tested positive for lung (NKX2.1) epithelial marker, pan-epithelial marker (E-CAD), AT2 cells (SPC^+^), and AT1 cells (PDPN^+^ or AQP5^+^). They also found that ESC-derived alveolar organoids exhibited high expression levels of ACE2 and TMPRSS2, and only a subpopulation of AT2 cells (about 30–40%) was sensitive to SARS-CoV-2 infection [19].

**Table 3 biomedicines-11-03034-t003:** Organoid culture from ESC-derived AT2 cells.

Cell Source	Feeder Cell Type (Epithelial to Feeder Ratio)	Medium	Model
Mouse ESC			
Very small embryonic-like stem cells (VSELs) [80]	Mlg2908 (3:100)	DMEM/F12, FBS (10%, *v*/*v*), penicillin/streptomycin (0.5%, *v*/*v*), 1 M HEPES (0.1%, *v*/*v*), and ITS (1%, *v*/*v*)	VSELs produce functional BASC and AT2
Mouse fetal lung bud tip-derived PSC [82]	Feeder-free	Medium including B27 supplement, BSA, FGF-7, all trans retinoic acid, and CHIR-99021	Lung developmental model
EPCAM + embryonic (e16.5) bi-progenitor cells [2]	Feeder-free	DMEM/F12, 1%FBS, and Fgf-7	Fgf signaling regulates alveolar fate
Human ESC			
hESCs-derived AT2 cells [19]	unknown	Ham’s F12 dexamethasone (8-Br-cAMP (3-isobutyl-1-methylxanthine) KGF, B-27, BSA, ITS premix, CHIR99021 and SB431542	SARS-CoV-2 infection model
Human fetal lung-bud-tip-derived PSC [82]	Feeder-free	Medium including B27 supplement, BSA, FGF-7, all trans retinoic acid, and CHIR-99021	Lung developmental model
Human embryonic lung tips and stalks [33]	PDGFRB^+^ lung embryonic mesenchyme	rhEGF, rhNoggin, rhFGF-10, rhFGF-7, CHIR99021, RSPO-1, and SB431542	Lung developmental model
Fetal lung bud tip progenitor cells [81]	Mesenchymal cells	B27 supplement, 0.05% BSA, FGF-7, FGF-10, BMP4, all trans retinoic acid, and CHIR-9902	Lung developmental and disease models

### 4.2. Feeder-Free Organoid Culture Systems

Although AT2 organoid co-cultures have been used to model lung diseases, they have several drawbacks that render them an undesirable system. One major drawback is the separation of AT2 cells from feeder cells for subsequent transplantation, as transplanting fibroblast cells can induce fibrosis following injury [83,84]. The variations in the type of feeder cells and the composition and concentration of growth factors secreted by the lung mesenchyme make co-culture disadvantageous for modeling cellular therapies. The co-culture system’s niche induces the differentiation of AT2 to AT1 cells. Recent studies have revealed that, in feeder-free organoid cultures, both AT2 cells and iAT2 cells do not spontaneously differentiate into AT1 cells. Instead, their differentiation into AT1 cells necessitates the addition of specific AT1 differentiation induction factors [85]. The feeder-free organoid cultures of AT2 cells in defined medium conditions (AT2 maintenance medium) maintain AT2 cells in a proliferative state and can be scaled up and passaged multiple times to yield a large number of AT2 cells. Feeder-free, serum-free cultures maintain the clonal expansion of EpCAM^+^ LysoTracker^+^ AT2 cells for up to 180 days [56]. These AT2 cells can be stored in liquid nitrogen for extended periods, and the same passage can be used for multiple experiments. This approach increases the reproducibility of results and reduces the number of animals used in the research for isolating AT2 cells, as the cells isolated from one mouse can be used for multiple experiments or studies.

Kastura et al. reported a feeder-free, scalable, chemically defined, and modular alveolosphere culture system for the propagation and differentiation of human alveolar type 2 cells derived from human lung tissues [9]. With an understanding of the niche signals required by AT2 cells for spheroid formation, researchers used defined media and added growth factors like Wnt pathway modulators, BIRB, DMH, EFG, FGF10, and Noggin to support the proliferation and self-renewal of AT2 cells without co-culturing with feeder cells. The authors employed single-cell RNA sequencing and analyzed the differential expression of ligand-receptor pairs on AT2 and fibroblast cells to develop and optimize the feeder-free and serum-free culture media for mouse and human AT2 alveolosphere expansion, maintenance, and differentiation. Based on this analysis, they tested different combinations of ligands and molecules to formulate a serum-free and feeder-free (SFFF) medium (Table 4). They observed that IL-1β could increase organoid size without affecting colony numbers. They also added Noggin and DMH1 to SFFF medium to inhibit BMP signaling, promoting a higher proportion of AT2s in the alveolosphere. BMP signaling has been reported to induce AT2 differentiation.

Salahuddeen et al. [67] reported long-term, chemically defined, feeder-free organoid cultures for AT2 cells and KRT5^+^ basal cells. Wnt5a signaling has been shown to play an important role in maintaining the stemness of mouse AT2 cells [58]. The inhibition of Wnt signaling decreases organoid formation by inhibiting AT2 self-renewal and initiating AT1 differentiation [65]. Recently, Salahudeen et al. [67] showed that an EGF/NOGGIN-containing defined medium was sufficient for the clonal proliferation of AT2 organoids, which was attenuated by blocking global endogenous Wnt biosynthesis, consistent with previous studies showing the requirement for Wnt signaling in mouse AT2 cells [58]. This defined medium supported AT2 proliferation, long-term propagation, multiple passaging, and differentiation into AT1 cells. It is well-established that Fgfr2 signaling plays a crucial role in regulating embryonic lung development. The downregulation of Fgfr2 signaling has been shown to inhibit embryonic airway branching [88,91,92]. Recent research has highlighted the importance of Fgfr2 signaling in the lifelong maintenance of AT2 cells. During juvenile stages, it prevents the reprogramming of AT2 cells into AT1 cells, and in adulthood, it suppresses apoptosis in AT2 cells. The Fgfr2 receptors are initially expressed on bipotent AT2 progenitor cells and continue to be expressed on mature AT2 cells. Stromal cells express growth factors Fgf7 and Fgf10, which bind to the Fgfr2 receptor, initiating Fgf signaling. Douglas et al. demonstrated that culturing bipotent progenitor cells for 4 days in DMEM/F12 with Fgf7 (at 50 ng/mL) and 1% FBS in Matrigel resulted in the formation of large alveolospheres containing differentiated Sftpc^+^ Rage^−^ AT2 cells and squamous Sftpc^−^ Rage^+^ Pdpn^+^ AT1 cells. In contrast, the absence of Fgf7 or treatment with the Fgfr inhibitor FIIN-1 led to epithelial progenitor cells failing to differentiate into AT1 or AT2 cells and prevented organoids’ development [2].

### 4.3. 3D Matrix Alternatives to Matrigel for Organoid Cultures

Organoid growth not only depends on growth factors and cell–cell interactions but also on signals from the extracellular matrix (ECM), which is an intriguing aspect of stem cell niches. Accumulated evidence has suggested that both physiological and mechanical cues from the extracellular matrix (ECM) may contribute to the maturation and differentiation of type 2 alveolar epithelial cells (AT2) [85]. In co-culture experiments involving iAT2 and fibroblasts, it was observed that fibroblasts express ECM genes and growth factors when AT2 cells are undergoing a transition or transdifferentiation into AT1 cells [54]. To date, both co-cultures and feeder-free 3D cultures of AT2 organoids have primarily relied on Matrigel [4,9,54]. Matrigel is the most commonly used medium to support 3D organoid growth. It is an ECM derived from cancerous mouse tissue and is commonly utilized for stem and cancer cell proliferation. While organoids grown in Matrigel mimic tissue structure, organ physiology, and function, it is derived from animal sources and presents a significant challenge in defining culture conditions due to lot-to-lot variability. The Matrigel system also produces heterogeneous organoids in terms of shape, size, and composition [79]. A chemically defined matrix system would provide more reliable data that could be clinically translated. Hoffman et al. conducted a study on a hydrogel derived from the extracellular matrix (ECM) of human alveolar cells (referred to as aECM hydrogel) for 3D organoid culture [54]. This hydrogel was obtained by processing alveolar-enriched fractions of decellularized human lungs. Their proteomics analysis revealed, that while the hydrogel did not encompass the complete spectrum of native ECM proteins, it exhibited enrichment in key proteins like COL1, COL3, and FBN1. The study demonstrated that the aECM hydrogel not only supported the proliferation of iAT2 (type 2 alveolar epithelial) cells and the formation of alveolospheres but also played a role in promoting the morphological differentiation of a subset of iAT2 cells into structures resembling human AT1-like cells. Additionally, the researchers investigated the influence of matrix stiffness by using different concentrations of aECM. Intriguingly, the higher stiffness of the aECM hydrogel led to a decrease in the SFTPC expression in iAT2 cells compared to hydrogels with less stiffness. Moreover, the bulk RNA sequencing of iAT2 cells cultured in the aECM hydrogel revealed changes in the expression of genes associated with iAT2 maturation, transitional cell states, and AT1-associated markers. These findings suggest that ECM stiffness may play a significant role in directing cellular differentiation processes [54]. Another study reported the use of hyaluronic acid hydrogels for the generation and expansion of lung AT2 organoids. In this study, synthetic hyaluronic acid hydrogels with defined chemical and physical properties were used to grow and expand iPSC-derived AT2 organoids. The predefined microstructure of these hydrogels reduced the heterogeneity of organoid size and structure when compared to a 3D culture in Matrigel. This hydrogel system could maintain the fate of iPSC-derived AT2 cells and provide a straightforward and defined culture system for growing 3D organoids from primary mouse lung AT2 cells and other epithelial progenitor and stem cell aggregates [79].

## 5. Applications

### 5.1. Modeling Lung Diseases with Organoids

The alveolar organoid model has helped researchers understand the pathophysiology of SARS-CoV-2 within the lungs [9]. Human embryonic stem-cell-derived airway and alveolar 3D organoid models have been used to study the infection and spread route of SARS-CoV-2 within the lungs. These models provide insights into the viral cell tropism and early cell response to viral infection, and they present a platform for testing new drugs to treat COVID-19 [19]. In another study [67], a feeder-free AT2 organoid was used to model COVID-19-associated pneumonia. This model allowed for ACE2 expression on the exposed apical surface, enabled the infection of AT2 cultures with SARS-CoV-2, and identified club cells as a target population.

Lung cancer is the leading cause of cancer-related deaths globally. The absence of appropriate ex vivo models of the human alveolar epithelium has hindered our understanding of lung cancer pathogenesis and related therapy development. Early-stage diagnosis and treatment are essential for preventing cancer relapse and saving lives. Recently, Dost et al. [68] used both mouse AT2 and human iPSCAT2 organoid models to uncover the early consequences of oncogenic KRAS expression in vivo. Their work has provided novel tools for extensive data collection and studying the transcriptional and proteomic changes that distinguish normal epithelial progenitor cells from early-stage lung cancer. Their study revealed that reductions in AT2 lineage marker gene expression are an early consequence of oncogenic KRAS. Multiomics studies demonstrated that SPC-high cells in Kras activation and p53 loss (KP) lung tumor organoids exhibit higher tumorigenic capacity in the lung microenvironment compared to Hmga2-high cells [93]. Additionally, alveolar organoids derived from patient-specific iPSCs have been used in modeling and understanding the pathogenesis of Hermansky–Pudlak-syndrome-associated interstitial pneumonia (HPSIP). Proteomic analysis revealed abnormal membrane trafficking and mitochondrial function in HPS1 patient-specific alveolar organoids [59].

### 5.2. Lung Developmental Studies

HESC-derived organoids have been used to model human lung development. These organoids contained early-stage proximal and distal airway epithelial cells, including early-staged alveolar type 2 (AT2) cells (SPC^+^/SOX9^+^) and immature alveolar type 1 (AT1) cells (HOPX^+^/SOX9^+^) in vitro. However, when transplanted in vivo for the short term, these organoids differentiated into only a few distal progenitor epithelial cells (NKX2.1^+^, SOX9^+^, and P63^+^). In contrast, the long-term transplantation of these organoids resulted in the differentiation of lung distal bipotent progenitor cells (PDPN^+^/SPC^+^/SOX9^+^), AT2 cells (SPC^+^ SPB^+^), and immature AT1 cells (PDPN^+^, AQP5^−^). These long-term transplanted organoids also contained other cell types present in lung tissues, such as mesenchymal cells, vasculature, neuroendocrine-like cells, and nerve fiber structures [94].

### 5.3. Drug Discovery/Target Screening

In a recent study, mouse primary AT2 and human iPSCs-derived AT2 organoids were used to investigate the early stages of lung adenocarcinoma (LUAD) driven by KRAS mutation. The data from the alveolar organoids model may be useful in screening novel drug targets and developing new drug molecules to prevent lung cancer growth at the early stage. Another study used hESCs-derived airway and alveolar lung organoids to show that Remdesivir and an antibody potentially inhibited SARS-CoV-2 replication in lung organoids. Recently, Zhao et al. used hESCs-derived airway and an alveolar lung organoid model to study the life cycle and target cell types of human adenoviruses type 3 (HAdV-3) and type 55 (HAdV-55). The results showed that HAdV-55 had a higher replication efficiency and was more infectious than HAdV-3, infecting both basal and AT2 cells. They found that the antiviral drug cidofovir had low cytotoxicity to organoids and exhibited good antiviral effects against HAdV-3 and HAdV-55 [19,53,68].

### 5.4. Preclinical Cell Therapy

Feeder-free organoid-derived AT2 cells have shown great potential for cellular therapy in lung regeneration. In a recent study, mesenchyme-free AT2 organoids were transplanted into the lungs of mice injured by influenza. The transplanted organoids retained their AT2 fate; however, in some cases, they adopted a dysplastic fate. These dysplastic organoids did not appear to improve the oxygen-exchange capability of the injured lungs in recipient mice. Further investigations have been requested to understand the molecular changes that occur in AT2 organoids after transplantation in the influenza-injured microenvironment in order to optimize organoid transplants [34].

### 5.5. Drug Toxicity Screening

In vitro cell-based models are indispensable tools for assessing the toxicity of new investigational drugs before pre-clinical testing. Several studies have shown that drugs that were tested as safe in animal models failed clinical trials and caused life-threatening toxicity, multiorgan failure, and death in humans [95,96,97]. Therefore, to prevent risking human life, human-cell-derived in vitro screening models should be used to recapitulate human biology and physiological characteristics much more closely than animal models. Human iPSCs may provide a continuous source of AT2 cells for growing organoids for drug toxicity screening [98]. The feeder-free model, given its ability to maintain AT2 cells for longer periods without undergoing AT1 differentiation, provides a platform for testing drug toxicity to AT2 cells, or a co-culture model may be employed for drug toxicity testing in the presence of endothelial, fibroblast, and AT1 cells. Yamamoto et al. demonstrated the applicability of hiPSC-derived alveolar organoids in drug toxicology studies, successfully recapitulating AT2-cell-specific phenotypes [43]. In another study, alveolar organoids were used to demonstrate the effectiveness of cidofovir against HAdV-3 and HAdV-55 infection with low cytotoxicity [53].

### 5.6. Cancer Models

The presence of significant genetic and phenotypic diversity among individuals with lung cancer underscores the demand for personalized medicine. In a study, researchers collected small samples of primary lung tumor tissue from five different histological subtypes of lung cancer. These tissue samples were used as in vitro models to represent each individual patient in the study. Under 3D culture conditions, the cancer cells proliferated and self-organized into spherical organoid structures within the Matrigel. Analysis showed that the lung cancer organoids histologically recapitulated features such as glandular patterns of the original adenocarcinoma tissues, including the expression of lung cancer markers. Xenograft tumors formed successfully by transplanting organoids subcutaneously into mice, demonstrating that the cancer organoids maintained tumor-initiating potential. The tumor-derived organoids could also be used for in vitro drug screening, with responses dependent on genetic alterations in the parental tumor [93]. Similarly, in another study, surgically resected tumors from patients with stage I/II non-small cell lung cancer (NSCLC) were obtained. These tumors were successfully cultured as patient-derived tumor spheroids (PDS) in a 3D system for over 120 days. The PDS cultures showed a 100% success rate, and the cells within these spheroids exhibited characteristics consistent with their cancer type. Overall, the study’s findings support the creation of an expandable 3D in vitro NSCLC model for drug screening and long-term studies, including the development of drug-resistant models [99]. However, organoid models derived from lung tumor cell lines or from lung tumors themselves are not well-suited to model the events in early-stage tumorigenesis [35,100,101].

## 6. Limitations and Future Directions

In the last decade, 3D organoid modeling has provided excellent tools for studying human diseases’ pathology, organ development, and platforms for drug testing. The many potential applications of this technology are only beginning to be explored [23]. Organoids grown in a defined culture system can be used for transplantation studies. A completely defined system for human alveolospheres must be free of any animal components so that the data generated from research using these systems can be directly applied to clinical studies. A synthetic hydrogel system combined with a defined, animal-component-free culture medium could provide such a system. Such species-specific defined conditions are needed to study the effects on mice or human AT2 cells and for the high-throughput screening of pharmaceutical molecules and genomic studies to discover drugs to treat lung diseases. However, there are some limitations. AT2 organoids underrepresent the lung structure in terms of cellular complexity, cell arrangement, and protein expression. The mechanical forces produced at the time of breathing cannot be studied in the organoid model. Compared with polarized tight monolayers cultured at the air–liquid interface, AT2 organoids cannot be used to study barrier functions, including transepithelial ion transport, permeability, polarization, and cilia movement. AT2 organoids were filled with a culture medium, and the cells were not polarized with apical and basolateral compartments [102,103]. Recently, primary AT2 cells derived from patients have been successfully expanded within feeder-free alveolar organoids. These cells have been employed in the development of a lung-on-chip model mimicking the alveolar structure [36]. This chip is supported by a perfusion system that continuously maintains the flow of a nutrient medium through the channels on both sides of the epithelial cells layer, and the shear force generated by fluid flow stretches and relaxes the AT2 epithelial cells, providing in-vivo-like physiologic conditions. These mechanical cues have significant importance in defining cellular fate, migration, differentiation, disease progression, and lung development. In chronic end-stage lung diseases, lung transplantation is the only option to save the life of a patient. But the ratio of the demand to donor lung tissue makes this option unviable [104]. Scaling up organoid culture using well-defined animal-component-free advance conditions may provide a solution for developing clinical-grade lung tissue grafts for transplantation in end-stage lung diseases.

## 7. Conclusions

Three-dimensional organoid technology has opened new avenues for regenerative medicine and provided a platform for novel drug screening, as well as for developing diagnostics in combination with editing technology for gene therapy and tissue engineering. In addition, patient-derived organoids have given scientists a new tool to predict drug responses in a personalized fashion. The methods discussed in this review article can help researchers overcome the limitations of current approaches to modeling human alveoli, and they should be useful for disease modeling and regenerative medicine. Despite a few shortcomings of the organoid system, it has proven to be a valuable tool in studying lung diseases, including the recent COVID-19 pandemic. The 3D AT2 organoid system has been used to elucidate many gene functions over the years, although new approaches may be required to fully define this powerful 3D model system. 

## Figures and Tables

**Figure 1 biomedicines-11-03034-f001:**
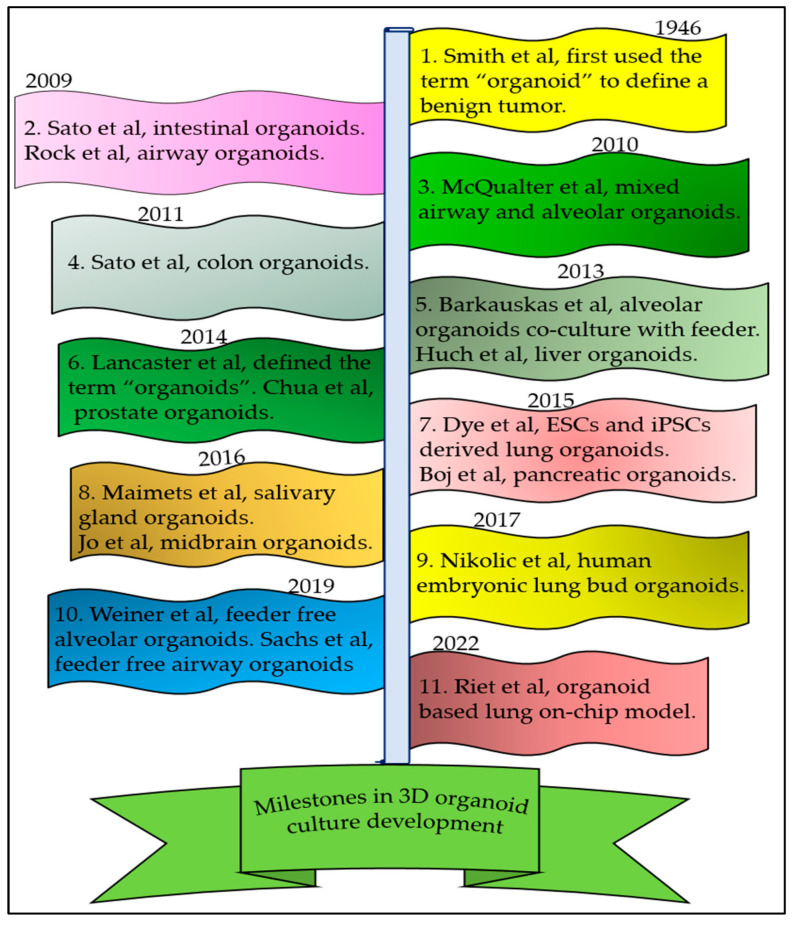
The milestones in 3D alveolar organoid development [4,20,21,22,24,29,31,32,33,34,36].

**Figure 2 biomedicines-11-03034-f002:**
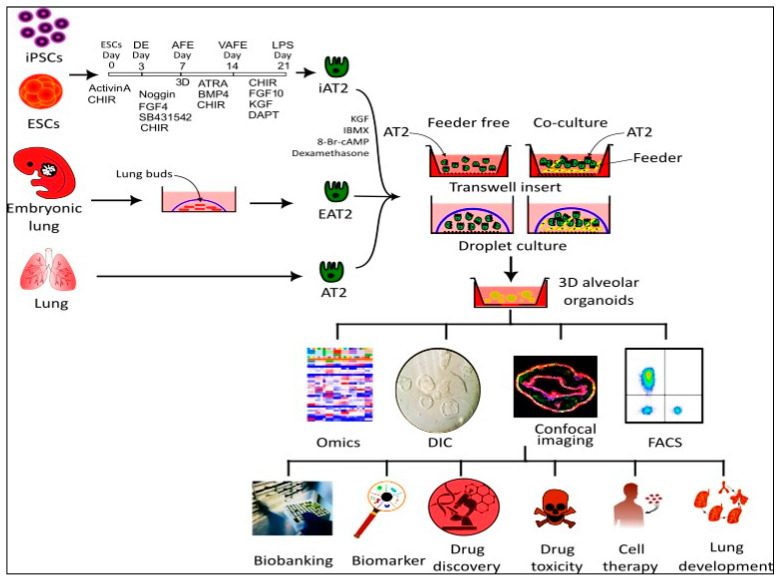
Overview of AT2 organoid cultures. AT2 cells can be derived from primary lung tissues, ESCs, embryonic lung buds, or iPSC lines. EPCAM^+^ rodent or HTII-280^+^ human AT2 cells are sorted via MACS or FACS for growing organoids. AT2 cells suspended in Matrigel with or without feeder cells are planted in transwells or as drops into six well plates or petri dishes. Organoids can be analyzed for colony-forming efficiency, morphology and structure, cell lineages, omics, and the development of disease models. In addition, organoids have been used for lung development studies, drug screenings, and transplantation investigations. iAT2: induced AT2 cells; EAT2: embryonic lung derived AT2 cells; DIC: differential interference contrast.

**Table 4 biomedicines-11-03034-t004:** Cytokines and growth factors in defined culture medium.

Growth Factor/Component	Function
SB431542 [72]	TGF-β inhibitor; the TGF-β signaling pathway is a key regulator of the epithelial-supportive capacity of lung stromal cells
CHIR99021 [32]	Wnt pathway activator/(GSK) 3 inhibitor, inhibiting both GSK3β (IC₅₀ = 6.7 nM) and GSK3α (IC₅₀ = 10 nM)
BIRB796	Potent inhibitor of p38 MAPK (Kd = 0.1 nM)/At 10 μM; BIRB-796 can also inhibit JNK2α2
R-spondin2 [86]	A co-activator of Wnt/β-catenin signaling; plays an important role in embryonic lung development and adult lung homeostasis and regeneration
DMH-1 (dorsomorphin homolog 1)	Inhibitor of activin receptor-like kinase 2 (ALK2; IC₅₀ = 13–10^8^ nM)
N-AcetylL-Cysteine	Antioxidant
EGF [87]	Regulates cell cycle, proliferation, and developmental processes; promotes regenerative alveolarization
HGF [31]	Regulates epithelial proliferation and lineage commitment
FGF10 [31,70]	Regulates epithelial proliferation and lineage commitment
FGF-7 (KGF) [2,88]	Stimulates the growth of AT2 organoids through fgfr2 signaling
Heparin [89]	Activates Wnt signaling
B-27 supplement	Serum-free supplement used to support the low- or high-density growth and short- or long-term viability of cells
Noggin [32]	BMP signaling inhibitors
IL-1β (0–4 days) [49]	IL-1β signaling directly promotes the reprogramming of AT2 Cells/ IL-1β treatment increased organoid size and formation efficiency
Y-27632 (0–4 days) [90]	RHO/ROCK pathway inhibitor
ITS	Insulin promotes glucose and amino acid uptake, lipogenesis, intracellular transport, and the synthesis of proteins and nucleic acids; transferrin is an iron carrier, and it may also help to reduce toxic levels of oxygen radicals and peroxide; selenium, as sodium selenite, is a co-factor for glutathione peroxidase and other proteins and is used as an antioxidant in media
N2 supplement	Serum-free supplement based on Bottenstein’s N-1 formulation; it is recommended for the growth and expression of neuroblastomas

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
