# Peer review of "Alveolar Type 2 Epithelial Cell Organoids: Focus on Culture Methods"

_biomedicines, 2023, doi:10.3390/biomedicines11113034_

Round 1
Reviewer 1 Report
Comments and Suggestions for Authors
In this review article, the authors conduct a comprehensive review of the cultivation of Alveolar type 2 epithelial cell organoids. The context provides detailed insights into cultivation methods and applications. I believe that readers will gain a deeper understanding of the cultivation of Alveolar type 2 epithelial cell organoids via this article. Here are some suggestions.
- Please providing additional information in the Introduction section to explain why 3D organoids on AT2 is of paramount importance. Could AT2 organoids increase cell proliferation and expression of tissue-specific functions?
- On Line 144, please remove the period before the reference numbers [47-49].
- Please consider relocating Figure 2 to the end of Section 4 “AT2 organoid culture methods”.
- Move Table 4 to the end of Section 4.2, after the description of feeder-free organoid culture systems.
Reviewer 2 Report
Comments and Suggestions for Authors
The review by Jain et al provides a very comprehensive and thorough evaluation on the various approaches used in generation of AT2 organoids, and discuss the merits and demerits of the culture system. Although this model system lacks many factors in the tissue/tumor microenvironment, it is the closest in vitro system that resembles the in vivo conditions, and is the most valuable model for toxicity testing, drug screening, development of personalized medicine etc. As such, it is of immense value to the scientific field. Thus, an in-depth analysis of the conditions that allow the perfect growth of the organoids is of significant importance and the authors have done an excellent job in summarizing the behaviour of organoids under various culture conditions, which would be of value to those conducting studies with such systems. This reviewer has no specific/additional comments.
